# Trends in maternal mental health during the COVID-19 pandemic–evidence from Zambia

Irene Falgas-Bague[1,2,3]*, Thandiwe Thembo[4], Jeanette L. Kaiser[5], Davidson H. Hamer[5], Nancy A. Scott[5], Thandiwe Ngoma[5], Ravi Paul[4], Allison Juntunen[5], Peter C. Rockers[5], Günther Fink[1]

1 Department of Epidemiology and Public Health, Swiss Tropical and Public Health Institute, Basel, Switzerland, 2 Department of Medicine, Harvard Medical School, Boston, Massachusetts, United States of America, 3 Department of Medicine, Mongan Research Institute, Disparities Research Unit, Massachusetts General Hospital, Boston, Massachusetts, United States of America, 4 Department of Psychiatry, University of Zambia, Lusaka, Zambia, 5 Department of Global Health, Boston University School of Public Health, Boston, Massachusetts, United States of America

* Irene.falgasbague@swisstph.ch

## Abstract

The COVID-19 pandemic has increased social and emotional stressors globally, increasing mental health concerns and the risk of psychiatric illness worldwide. To date, relatively little is known about the impact of the pandemic on vulnerable groups such as women and children in low-resourced settings who generally have limited access to mental health care. We explore two rounds of data collected as part of an ongoing trial of early childhood development to assess mental health distress among mothers of children under 5-years-old living in two rural areas of Zambia during the COVID-19 pandemic. We examined the prevalence of mental health distress among a cohort of 1105 mothers using the World Health Organization's Self-Reporting Questionnaire (SRQ-20) before the onset of the COVID-19 pandemic in August 2019 and after the first two infection waves in October-November 2021. Our primary outcome was mental health distress, defined as SRQ-20 score above 7. We analyzed social, economic and family level characteristics as factors modifying to the COVID-19 induced changes in the mental health status. At baseline, 22.5% of women were in mental health distress. The odds of mental health distress among women increased marginally over the first two waves of the pandemic (aOR1.22, CI 0.99–1.49). Women under age 30, with lower educational background, with less than three children, and those living in Eastern Province (compared to Southern Province) of Zambia, were found to be at highest risk of mental health deterioration during the pandemic. Our findings suggest that the prevalence of mental health distress is high in this population and has further worsened during COVID-19 pandemic. Public health interventions targeting mothers' mental health in low resource settings may want to particularly focus on young mothers with limited educational attainment.

**Data Availability Statement:** ***PA AT ACCEPT: Please follow up with the authors for data available at accept. *** We will submit the full dataset to replicate the results upon acceptance of the article.

**Funding:** IFB, RP, TT, received no specific funding for this work For NAS, DHH, JLK, AJ, PR, GF, this work was supported by Grand Challenges Canada (TTS-1802-21377 and TTS-2009-35996), and the United States Agency for International Development (USAID; grant number 72061119FA00001). The funders had no role in study design, data collection, analysis, decision to publish, or preparation of this manuscript. The content is solely the responsibility of the authors and does not reflect positions or policies of Grand Challenges Canada or USAID.

**Competing interests:** The authors have declared that no competing interests exist.

## Introduction

The outbreak of COVID-19 has changed daily routines and lives of all segments of society worldwide [1]. The impact of the pandemic goes well beyond the direct consequences of the viral infection, adversely affecting the global and domestic economy, the environment, local living conditions and the incidence and course of numerous chronic conditions [2–5]. Measures taken by governments around the world to reduce the spread of the virus included national lockdowns, as well as limiting access to usual sources of support and income such as schools and market places [6]. These measures resulted in increased burdens of existing family struggles and are shown to have affected women more than men [7]. Even though pandemic-related stressors have affected the entire population, the impact of COVID-19 pandemic appears particularly high among women, with increases in the incidence of gender-based violence, a worsening gender gap due to lost employment and career opportunities, rising levels of anxiety and depression, and an increased risk of adverse pregnancy and birth outcomes [8,9]. For women living in low-and-middle income countries (LMICs), and more specifically in sub-Saharan Africa, these negative consequences might be even more severe due to lack of social security benefits, poorer access to care, and restricted access to income from self-employment [10,11]. In addition, many maternal and child healthcare services were de-prioritized [12] within strained health systems [13] and moved to virtual delivery platforms even when systems to support these platforms were still not in-place. This often resulted in missed appointments, late diagnoses and higher morbidity and mortality rates [14]. Despite all of these challenges, surprisingly little has been published on the actual impact of the pandemic on women's mental health in sub-Saharan Africa.

In Zambia, similar to other LMICs, COVID-related public health measures included the closure of restaurants, companies, primary schools and colleges, as well as reduced in-person health care services [15]. Companies closed which led to layoffs of many domestic workers. Lack of strong social protection measures compromised the already impoverished household economies [16–18], resulting in even more severe poverty, high food insecurity, housing instability and limited access to healthcare [19,20]. Two recent studies evaluated women's mental health during the COVID-19 pandemic in Zambia. One showed high levels (above 40%) of stress and anxiety among health professionals [21] and a qualitative study exposed widespread perceptions of reduced access to healthcare due to lack of transportation during lockdowns and increased stress among women related to increased domestic violence events, loss of employment, childcare demands, and family isolation [22]. However, there is currently no evidence on changes in mental health distress among women in the country from before to during the COVID-19 pandemic.

Psychological distress, in general, and, especially in mothers of young children, negatively influences the usual way of thinking. It is associated with feelings such as anxiety, anger and/or sadness and affects usual behaviors by changing sleep patterns, diet, or interpersonal relationships. Overall, it leads to a reduced functioning that, for women in charge of young children, may encompass long-lasting impacts not only for themselves but also to their descendants [23,24]. Maternal and perinatal stress has been associated with negative child developmental outcomes such as psychological and behavioral dysregulation, dermatologic pathology and children's obesity, among others [25]. Further, in sub-Saharan Africa, underlying contextual factors have been found to aggravate the risk of symptoms of depression and anxiety among women [11,26].

Here, we aim to assess mental health distress among one of the most vulnerable populations, mothers with children under 5 years old, living in two different rural and semi-urban areas of Zambia. Using data from a cohort of 1105 women who had their babies in 2019 and

were re-assessed after 2 years, we aimed to describe and compare the mental health burden prior to and during the COVID-19 pandemic in rural Zambia, which is likely representative of many parts of sub-Saharan Africa.

## Methods

This is a secondary analysis of data collected within a clustered randomized clinical trial (NCT03991182) assessing the impact of community-based parenting groups on child developmental outcomes. Ethical approval was granted from two academic institutions (Boston University and University of Zambia). For this study, we used data collected during the initial baseline assessment in August-September 2019 as well as during the endline visit in October-November 2021. Data were collected using SurveyCTO® Collect Software (Dobility, Inc, Cambridge, MA), downloaded and securely stored at the United States-based academic institution. Written informed consent was obtained from all the participants included in the study.

### Study participants and site

Recruitment followed a multistage random sampling procedure to select a representative sample of women living within the surrounding areas of ten purposively selected health facilities from four districts across Southern and Eastern Provinces of Zambia (Nyimba, Choma, Pemba, and Kalomo districts). Trained bilingual (English and local language) interviewers visited all households with children born during the prior year in the catchment areas of these ten facilities with the support of community volunteers. At baseline, women above 15 years of age were screened for eligibility and all consenting mothers with infants up to 5 months old enrolled in the study. These same mother-child dyads were followed up at endline two years later. Pemba and Choma districts were grouped together in the last census in 2010. The districts have a population of 247,860 and a population density of 34/km², with 68.7% of its population being rural. Kalomo district has a population of 258,570 and a population density of 17.2/km², with 91.8% of its population living in rural areas [27,28]. Nyimba district has a population of 85,025 and a population density of 8.1/km², with 91% of its population living in rural areas [29]. The populations of these districts have a generally low socio-economic status, with limited access to improved sources of electricity, water or sanitation, and low educational attainment. Farming and selling farming products are the primary sources of income in all three districts [30].

### Measures

Our primary outcome of interest was women's mental health distress. Women's mental health was assessed using the Self-Reporting Questionnaire (SRQ-20), a 20-item retrospective survey with yes/no questions focusing on mental health distress in the 30 days preceding the interview [31]. The SRQ-20 was designed by the World Health Organization (WHO) and is a relatively easy-to-administer population-level screening device for common mental disorders [31] including non-specific forms of depression, anxiety and post-traumatic stress disorder (PTSD) related symptomatology. The SRQ-20 includes both somatic items (e.g., headaches, loss of appetite, tiredness) and psychological items (e.g., feeling unhappy, nervous, and worthless). It has been widely used in LMICs [32], including in many African settings [33,34] and specifically in Lusaka, Zambia [35] showing an acceptable internal consistency [36]. SRQ-20 scores are calculated by adding all the affirmative responses. Typically, a cut-off point to distinguish "cases" of poor mental health ranges from 7 to 8 in similar populations [32,35]. For this study, we followed existing literature [35] using SRQ-20 for assessing women's mental health in sub-Saharan African settings and used scores above 7 as a cut-off for identifying cases with

potential mental health distress. Trained interviewers administrated the SRQ-20 questionnaire in English and in the two main local languages spoken in the recruiting areas. We followed a process of formal translation and back translation to ensure consistency on the use of the local languages for the assessment.

## Covariates

We focused on age, educational attainment (less than primary school, primary school completion, and secondary-school completion), number of children in the respondents' care, marital status (married vs. non-married status), and household assets as primary covariates of interest. Household assets were combined into an index using principal component analysis and categorized into quintiles [37].

## Statistical analysis

We first generated descriptive statistics for the study population, and then computed the proportion of women with mental health distress by subgroup and survey round. We then pooled all data collected at baseline and endline and used logistic regression models to test for increasing mental health distress over time. The primary exposure variable of interest was an indicator variable for observations in the post-Covid period (coded to 1 for endline observations, 0 for baseline observations). To address confounding concerns due to changing sample characteristics, we estimated both adjusted and unadjusted models. We also estimated separate (stratified) trends for subgroups of interest. All analyses were conducted in Stata SE 16.

## Results

There were 1105 women assessed at baseline, of whom 942 (85%) were assessed again at endline. Out of the 1105 women assessed, 947 lived in the Southern Province districts of Choma/ Pemba and Kalomo, and 148 women lived in Zambia's Eastern Province. Most women (42.8%) were under 30 years old, married (75.7%), had minimal educational background with primary school degree (52.9%), and had an average of three children (under 18 years old) under their care. Table 1 shows the percentage of women with mental health distress (SRQ>7) by subgroup and survey round. At baseline, 22.5% of women were in mental health distress. Prevalence of mental health distress was higher among mothers over 40 years of age (39.2%).

Number of women as well as % of women with mental health distress (SRQ > 7) by survey round. Mental health scores are missing for seven women in 2019, and for three women in 2021.

As Table 1 shows, mental health distress increased in most subgroups. Between the pre- and post-COVID-19 timepoints, the largest increases in mental health distress were observed for women ages 20–29 years old (with a 29.4% increase in mental health distress), women with less than primary education degree (+52.4%), women with less than three children under their care (+38.4%), and women at the highest quintile of household assets (+28.9%). In Eastern Province, psychological distress doubled, increasing from 27.2% to 55.6%, a relative change of 102.8%.

Table 2 displays our overall trend estimates. Overall, the odds of mental health distress increased by a marginally significant 22% (OR 1.22 [CI 0.99–1.49] p = 0.085). Controlling for sociodemographic characteristics (age, educational background and marital status–Table 2, column 2) as well as site, number of children under their care, and household assets (Table 2, column 3) changed estimates only marginally.

In multivariate regression models, depicted in Table 3, the largest risk increases between the timepoints were found for women under 30 years of age (OR 1.29 [CI 1–1.67]), mothers

**Table 1. Mental health distress by subgroup and survey round (n = 1098).**

| | Prevalence of mental health distress (SRQ > 7) | | | | |
|---|---|---|---|---|---|
| | 2019 Survey | | 2021 Survey | | Relative change in % |
| | N | % | N | % | |
| Age in years | | | | | |
| < 20 | 309 | 18.4% | 253 | 20.6% | 11.4% |
| 20–29 | 467 | 20.6% | 391 | 26.6% | 29.4% |
| 30–39 | 271 | 27.3% | 246 | 27.2% | -0.3% |
| 40+ | 51 | 39.2% | 49 | 44.9% | 14.5% |
| Education Status | | | | | |
| No education | 70 | 28.6% | 62 | 43.5% | 52.4% |
| Primary education | 579 | 25.7% | 494 | 28.5% | 10.9% |
| Secondary education | 398 | 19.3% | 340 | 20.6% | 6.4% |
| Married/cohabiting | 829 | 24.1% | 726 | 27.0% | 11.9% |
| Primary caregiver to | | | | | |
| < 3 children | 493 | 16.8% | 399 | 23.3% | 38.4% |
| 3+ children | 605 | 27.1% | 540 | 28.1% | 3.8% |
| Wealth quintile | | | | | |
| First (lowest) | 233 | 25.8% | 182 | 28.6% | 11.0% |
| Second | 211 | 21.3% | 186 | 26.9% | 26.0% |
| Third | 237 | 22.8% | 215 | 24.2% | 6.1% |
| Fourth | 208 | 24.0% | 181 | 27.6% | 14.9% |
| Fifth (highest) | 209 | 18.2% | 175 | 23.4% | 28.9% |
| Province | | | | | |
| Eastern | 157 | 27.4% | 144 | 55.6% | 102.8% |
| Southern | 941 | 21.7% | 795 | 20.8% | -4.3% |
| Overall | 1098 | 22.5% | 939 | 26.1% | 16.0% |

with less than three children under their care (OR1.51 [CI 1.01–2.11]), and women living in Eastern Province (OR 3.79 [CI 2.278–6.319]).

**Table 2. Unadjusted and adjusted trends over time.**

| | SRQ > 7 Change (Post 2021) | | |
|---|---|---|---|
| | Model 1 | Model 2 | Model 3 |
| | Unadjusted | Adjusted for sociodemographic characteristics | Adjusted for sociodemographic plus site, assets and number of children |
| Worsening of symptoms to mental health distress | 1.216* | 1.207* | 1.201* |
| | (0.993–1.490) | (0.982–1.483) | (0.975–1.481) |

Robust confidence interval form in parentheses:

*** p<0.01,

** p<0.05,

* p<0.1.

Sociodemographic characteristics: Age, educational attainment, marital status.

**Table 3. Change in mental health distress by subgroup (stratified results).**

|              | Age < 30 | Age 30+ | No education | Primary or more | < 3 children | 3+ children | Southern Province | Eastern Province |
|--------------|----------|---------|--------------|-----------------|--------------|-------------|-------------------|------------------|
| Post (2021)  | 1.289*   | 1.039   | 1.934        | 1.156           | 1.507**      | 1.044       | 0.933             | 3.794***         |
|              | (0.996–1.667) | (0.723–1.493) | (0.861–4.344) | (0.929–1.439) | (1.075–2.113) | (0.799–1.363) | (0.737–1.182) | (2.278–6.319) |
| Observations | 1,420    | 617     | 132          | 1,905           | 892          | 1,145       | 1,736             | 301              |

Table shows results of stratified logistic regression models. Displayed coefficients are odds ratios with 95% confidence intervals in parentheses. All models are fully adjusted (model specification 3 in Table 2)

*** p<0.01,

** p<0.05,

* p<0.1.

## Discussion

To our knowledge, this is the first quantitative study examining the trajectory of mental health distress of women with children under five during COVID-19 in sub-Saharan Africa. The study strongly highlights the high pre-COVID-19 levels of mental distress among mothers with children under five living in Zambia, as well as further deterioration in mental health during the pandemic. It also identifies mothers above 40 years old, mothers with lower levels of education and mothers with more than three children under their care as subgroups with particularly high levels of mental distress during the pandemic.

Our baseline data show higher levels of mental distress among women with lower educational background. Higher education attainment has been positively associated with mental health and wellbeing [38]. This association is important for LMICs and specifically in Zambia, where, in spite of the efforts to achieve universal education, current data show that still 17% of adult women did not complete primary school [39].

Although the study design does not allow causal interpretation of our estimates, it seems likely that the COVID-19 pandemic at least contributed to the increase of mental health deterioration. The almost uniform increase in mental health distress across all population groups is consistent with recent evidence from other LMICs highlighting the high prevalence of depression and anxiety symptoms across different regions of the world during the COVID-19 pandemic [40,41].

In our study, we found that younger mothers (less than 30 years old) and women with less than three children under their care were most strongly affected by deteriorating mental health scores during the COVID-19 pandemic. Qualitative research from Zimbabwe, Malawi and Zambia and quantitative research from South Africa have highlighted food insecurity, childcare responsibilities, increases in domestic violence events, and pre-existing health conditions as main factors for mental health distress during the pandemic in this setting [22,42–44]; it is possible that younger women as well as less educated women were disproportionally affected by these factors.

Increases in mental health distress during the pandemic were particularly noticeable in the Eastern region, where the proportion of women in mental health distress more than doubled over the study period. The Southern and Eastern provinces analyzed in this study differ in many respects, including socio-cultural characteristics, cultural values, and behaviors that may explain these differences. For instance, the three Southern districts in our sample have been affected by severe droughts and deforestation in the last decades leading to harder farming work and limited opportunities for self-maintenance for women compared to the Eastern province, where farming practices have been easier and women usually are able to rely on their own businesses. Another difference between the Eastern and Southern provinces is related to

polygamy practice, which is predominant in the South [45]. Overall, women living in the Southern Province districts may have been exposed to other adversity in recent years that may have increased their resilience and endurance when facing acute stressors, reducing the impact of COVID-19 public health measures, as it has been reported in other settings [46,47]. Further research will be needed to better understand these trends and dynamics and to identify other potential contributing factors.

Overall, our findings suggest that increases in mental health distress were large in subgroups with lower prevalence of mental health distress before the COVID-19 pandemic. These results differ from similar studies carried out in high and low-income settings, where lower mental health status at baseline predicted further mental health deterioration during the pandemic [44,48]. Research findings prior to COVID-19 pandemic, from similar settings to our study, show that women living in more challenging environments (i.e. polygamous marriages, long droughts, or who were in charge of more kids) appear to be better prepared to cope with new harsh realities such as the COVID-19 related social measures [46,49–51]. Public health authorities should consider the subgroup of young mothers as especially vulnerable to acute stress and prioritize the deployment of interventions to enhance the wellbeing of this population and, indirectly, prevent neurodevelopmental problems in future generations.

This study faces several limitations. First, the study was designed and launched before the COVID-19 outbreak, limiting the data collected on COVID-19, COVID-19-related government measures and other forms of socioeconomic impact of the pandemic. Second, our results might underestimate the true increases in mental health distress because baseline measures were taken when children were less than six months old, and some mothers were potentially still affected by postnatal depression. Similarly, other potential contributors to depression such as other co-occurring conditions affecting mothers (e.g. HIV new diagnosis) were not taken into account. Third, the pandemic and society's responses were frequently changing during this period and having more time points of assessment could have shed greater light on the mental health effects of that dynamism. Lastly, there is only limited evidence on the validity of the SRQ-20 in these settings. Even though the SRQ-20 is likely the most used tool for measuring mental health distress in the region and is thought to be acceptable [32], studies comparing its reliability and validity within the current context are needed to increase rigorous research in low-resourced settings.

## Conclusions

This study highlights the high burden of mental health distress among women with young children in Zambia, as well as the further deterioration of mental health during the pandemic. This deterioration was most pronounced among younger and less educated mothers, who likely should be prioritized when deploying mental health support interventions.

## Supporting information

**S1 File.**
(ZIP)

## Author Contributions

**Conceptualization:** Irene Falgas-Bague, Thandiwe Thembo, Nancy A. Scott, Thandiwe Ngoma, Ravi Paul, Peter C. Rockers, Günther Fink.

**Data curation:** Jeanette L. Kaiser, Nancy A. Scott, Thandiwe Ngoma, Allison Juntunen, Peter C. Rockers, Günther Fink.

**Formal analysis:** Irene Falgas-Bague, Günther Fink.

**Funding acquisition:** Nancy A. Scott, Thandiwe Ngoma, Günther Fink.

**Investigation:** Jeanette L. Kaiser, Nancy A. Scott, Thandiwe Ngoma, Allison Juntunen, Günther Fink.

**Methodology:** Günther Fink.

**Project administration:** Nancy A. Scott, Allison Juntunen.

**Resources:** Nancy A. Scott.

**Supervision:** Nancy A. Scott, Peter C. Rockers, Günther Fink.

**Validation:** Thandiwe Thembo, Davidson H. Hamer.

**Writing – original draft:** Irene Falgas-Bague, Thandiwe Thembo.

**Writing – review & editing:** Irene Falgas-Bague, Thandiwe Thembo, Jeanette L. Kaiser, Davidson H. Hamer, Nancy A. Scott, Thandiwe Ngoma, Ravi Paul, Allison Juntunen, Peter C. Rockers, Günther Fink.

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
