## [Decision Letter · Decision Letter 0]

22 Nov 2022

PONE-D-22-29001Trends in maternal mental health during the COVID-19 Pandemic – Evidence from ZambiaPLOS ONE

Dear Dr. Falgas-Bague,

Thank you for submitting your manuscript to PLOS ONE. After careful consideration, we feel that it has merit but does not fully meet PLOS ONE’s publication criteria as it currently stands. Therefore, we invite you to submit a revised version of the manuscript that addresses the points raised during the review process. The reviewers have provided thoughtful comments which you can find below. Please revise the manuscript as recommended by the reviewers. 

We look forward to receiving your revised manuscript.

Kind regards,

Ankit Jain, M.D.

Academic Editor

PLOS ONE

Journal Requirements:

Reviewers' comments:

Reviewer's Responses to Questions

**Comments to the Author**

1. Is the manuscript technically sound, and do the data support the conclusions?

Reviewer #1: Partly

Reviewer #2: Yes

Reviewer #3: Yes

Reviewer #4: Yes

2. Has the statistical analysis been performed appropriately and rigorously? 

Reviewer #1: Yes

Reviewer #2: Yes

Reviewer #3: I Don't Know

Reviewer #4: Yes

3. Have the authors made all data underlying the findings in their manuscript fully available?

Reviewer #1: Yes

Reviewer #2: No

Reviewer #3: Yes

Reviewer #4: Yes

4. Is the manuscript presented in an intelligible fashion and written in standard English?

Reviewer #1: Yes

Reviewer #2: Yes

Reviewer #3: Yes

Reviewer #4: Yes

5. Review Comments to the Author

Reviewer #1: This study evaluates trends in maternal mental health, pre and during the COVID-19 pandemic. The authors collected two rounds of data pre-COVID-19 in august 2019 and during the COVID-19 pandemic (October – November 2021) from women who have children under 5 years old living in two rural areas of Zambia. The primary outcome of the study was mental health distress, defined as World

Health Organization’s Self-Reporting Questionnaire (SRQ-20) score above seven. The secondary outcomes were analysis of social, economic, and family-level characteristics that may have influenced maternal mental health.

The study found that at baseline (Pre pandemic) 22.5% of women were in mental health distress, with a marginal increase in mental health distress over the two waves of the pandemic. The study concluded that women under age of thirty, with lower educational background, with less than three children, and those living in Eastern Province (compared to Southern Province), Zambia, were found to be at the highest risk of mental health deterioration during the pandemic

Strengths of the study:

Large sample size

This is the first quantitative study examining the trajectory of mental health distress in women with children under five during COVID-19 in sub-Saharan Africa.

Clarification needed:

Methodology:

The study did not mention how SRQ-20 was administered. It is a self-administered rating scale. Did authors require to translate this psychometric tool in the local language spoken by people in rural Zambia?

Participant recruitment: Authors did not mention how study participants were recruited. Did this study was advertised in the local media?

I would like to know about the participation of women who could not read and write and their ability to fill out SRQ-20.

How was participants' income calculated? How did authors stratify participants based on their income source?

How was data stored?

Reviewer #2: I have read with great interest the article titled 'Trends in maternal mental health during the COVID-19 Pandemic – Evidence from Zambia'. The authors have done a good job with the manuscript. While article does discuss about the maternal mental health issues faced during COVID, the article does have limitations. The authors have addressed few limitations but do not mention of possible co-morbid condition that have been proved to be a cause of depression. The study was conducted during the pandemic but does not discuss if the increase in depression rate was because of the pandemic. The increase in depression rate cannot be considered 'marginally significant' as the Confidence interval included 1.

The limitations may be edited to include these recommendations.

Reviewer #3: Thank you for shedding light on this topic. This paper has a remarkable potential to bring this issue into the mainstream through PLOS, but I suggest some minor revisions.

The paper needs proofreading for syntax and grammar.

The introduction is well-framed and discusses mental health issues prevalent in underserved population in terms of clinical presentation and social corelates of illness.

The discussion segment in paper could benefit from comparisons and contrasting with any other similarly designed international study by elaborating on cultural/international differences, if any, or lack thereof.

Reviewer #4: Authors have done this longitudinal study to study Trends in maternal mental health during the COVID-19 pandemic. Introduction is well written highlighting the importance of topic and lack of data about maternal health in local regions of Zambia pre and during pandemic. Introduction is easy to follow and makes a good argument to study the current topic. Methods are robust, well defined and clearly explained. Results are well explained and easy to understand for the readers. Tables are helpful. Discussion is well structured, highlights the importance of results. limitations are carefully examined and explained. Overall authors have done a great work in conducting this important study and presenting it in an intelligible way.

6. PLOS authors have the option to publish the peer review history of their article (what does this mean?). If published, this will include your full peer review and any attached files.

Reviewer #1: No

Reviewer #2: No

Reviewer #3: No

Reviewer #4: No

---

## [Author Response · Author response to Decision Letter 0]

21 Dec 2022

Reviewer 1 comments:

Strengths of the study:

Large sample size

This is the first quantitative study examining the trajectory of mental health distress in women with children under five during COVID-19 in sub-Saharan Africa.

Clarification needed:

Methodology:

1.1. The study did not mention how SRQ-20 was administered. It is a self-administered rating scale. Did authors require to translate this psychometric tool in the local language spoken by people in rural Zambia?

We thank the reviewer for this comment. The SRQ-20 was administrated by trained research interviewers making sure confidentiality was saved. SRQ-20 was translated and back translated in the two main local languages (Bemba and Tonga) spoken by people in rural Zambia. Quality assurance checks were done on a random basis to ensure fidelity to the assessments. We added information about the SRQ-20 administration on page 6 of the revised text, where we write:

“Trained interviewers administrated the SRQ-20 questionnaire in English and in the two main local languages spoken in the recruiting areas. We followed a process of formal translation and back translation to ensure consistency on the use of the local languages for the assessment.”

1.2. Participant recruitment: Authors did not mention how study participants were recruited. Did this study was advertised in the local media?

Apologies for the lack of information on recruitment. We used two stage cluster-random sampling for for recruiting a representative sample of women. 

In page 5 of the revised manuscript we added:

“Recruitment followed a multistage random sampling procedure to select a representative sample of women living within the surrounding areas of ten purposively selected health facilities from four districts across Southern and Eastern Provinces of Zambia (Nyimba, Choma, Pemba, and Kalomo districts). Trained bilingual (English and local language) interviewers visited all households with children born during the prior year in the catchment areas of these ten facilities with the support of community volunteers. At baseline, women above 15 years of age were screened for eligibility and all consenting mothers with infants up to 5 months old enrolled in the study. These same mother and child dyads were followed up at endline two years later.”

1.3. I would like to know about the participation of women who could not read and write and their ability to fill out SRQ-20.

Literacy was not an exclusion criteria for the participation in the study. SRQ-20 was administered by the trained interviewers in the woman’s local language. 

1.4. How was participants' income calculated? How did authors stratify participants based on their income source?

We thank the reviewer for noticing this lack of consistency on the naming of our socioeconomic variables. To assess economic status of household, we collected information on a range of household assets such as quality of construction materials in their household, availability of food and household supplies as a proxy of long-term income. These assets were then analyzed using principal component analysis following the methodology originally proposed by Filmer and Pritchett (2001)[1] and then used to categorize households into five asset quintiles. To increase clarity and consistency across the manuscript, we now replace “income” with “household assets” throughout.

1.5. How was data stored?

Data was captured electronically using SurveyCTO Collect Software installed on encrypted tablets. All data was transferred via a secure server managed by SurveyCTO and only accessed through an encryption key on staff computers. Data was downloaded and stored on a secure server at Boston University. We added the following in the revised manuscript page 5:

“Data were collected using SurveyCTO® Collect Software (Dobility, Inc, Cambridge, MA) , downloaded and securely stored at the United States-based academic institution.”

Reviewer #2: I have read with great interest the article […]. The authors have done a good job with the manuscript. While article does discuss about the maternal mental health issues faced during COVID, the article does have limitations. 

We thank the reviewer for the positive remarks.

2.1. The authors have addressed few limitations but a) do not mention of possible co-morbid condition that have been proved to be a cause of depression. B) The study was conducted during the pandemic but does not discuss if the increase in depression rate was because of the pandemic. C) The increase in depression rate cannot be considered 'marginally significant' as the Confidence interval included 1. The limitations may be edited to include these recommendations.

To respond to reviewer’s comments, we have expanded the limitation section. 

First, we added potential other explanations of our results such as other comorbid conditions associated with depression. In page 13 of the revised manuscript we added the following:

“Similarly, other potential contributors to depression such as other co-occurring conditions affecting mothers (e.g. HIV new diagnosis) were not taken into account.”

Second, we are aware that our analysis and results cannot infer causal associations between the pandemic and depression but only show an association between the events. We have clarified this concept on the discussion as well (page 11):

“Although the study design does not allow causal interpretation of our estimates, it seems likely that COVID-19 pandemic at least contributed to the increase of mental health deterioration.”

As for the mention to marginally significant results, we agree that this term is not uniformly defined in the literature. The confidence intervals we provide are based on a p-value of 0.05; our results have a p-value smaller than <0.1 which would be considered significant by many statisticians. Since this concept is not clear we think that marginally significant is the correct expression. To emphasize the statistic result, we have added the exact p-value (P-value = 0.085) to the results section. 

Reviewer #3: Thank you for shedding light on this topic. This paper has a remarkable potential to bring this issue into the mainstream through PLOS, but I suggest some minor revisions.

Thank you. We appreciate the positive remarks. 

3.1. The paper needs proofreading for syntax and grammar.

To respond the reviewer comments and improve the quality of the article, the authors’ team has reviewed the full manuscript and edited it for syntax and grammar. Edits can be found in the “track changes” version of the revised manuscript.

3.2. The discussion segment in paper could benefit from comparisons and contrasting with any other similarly designed international study by elaborating on cultural/international differences, if any, or lack thereof.

There are few articles using similar designs to assess mental health among women during COVID-19 pandemic. However, we expanded the discussion section with results from similar studies in different settings and added the following (page 11 and 12):

“Qualitative research from Zimbabwe, Malawi and Zambia and quantitative research from South Africa have highlighted food insecurity, childcare responsibilities, increases in domestic violence events, and pre-existing health conditions as main factors for mental health distress during the pandemic in this setting [2-5]; it is possible that younger women as well as less educated women were disproportionally affected by these factors. 

 “

“These results differ from similar studies carried out in high and low- income settings, where lower mental health status at baseline predicted further mental health deterioration during the pandemic.[5, 6] Research findings prior to COVID-19 pandemic, from similar settings to our study, show that women living in more challenging environments (i.e. polygamous marriages, long droughts, or who were in charge of more kids) appear to be better prepared to cope with new harsh realities such as the COVID-19 related social measures.[7-10]”

Reviewer #4: Authors have done this longitudinal study to study Trends in maternal mental health during the COVID-19 pandemic. Introduction is well written highlighting the importance of topic and lack of data about maternal health in local regions of Zambia pre and during pandemic. Introduction is easy to follow and makes a good argument to study the current topic. Methods are robust, well defined and clearly explained. Results are well explained and easy to understand for the readers. Tables are helpful. Discussion is well structured, highlights the importance of results. limitations are carefully examined and explained. Overall authors have done a great work in conducting this important study and presenting it in an intelligible way.

We want to thank the reviewer for the positive comments.

References

1. Filmer D, Pritchett LH. Estimating wealth effects without expenditure data—or tears: an application to educational enrollments in states of India. Demography. 2001;38(1):115-32.

2. Nyahunda L, Chibvura S, Tirivangasi HM. Social work practice: accounting for double injustices experienced by women under the confluence of Covid-19 pandemic and climate change impacts in Nyanga, Zimbabwe. Journal of Human Rights and Social Work. 2021;6(3):213-24.

3. Nyashanu M, Karonga T, North G, Mguni M, Nyashanu W. COVID-19 lockdown and mental health: Exploring triggers of mental health distress among women in the Copperbelt province, Zambia. International Journal of Mental Health. 2021:1-14.

4. Ahmed SA, Changole J, Wangamati CK. Impact of the COVID-19 pandemic on intimate partner violence in Sudan, Malawi and Kenya. Reproductive health. 2021;18(1):1-7.

5. Abrahams Z, Lund C. Food insecurity and common mental disorders in perinatal women living in low socio-economic settings in Cape Town, South Africa during the COVID-19 pandemic: a cohort study. Global Mental Health. 2022;9:49-60.

6. Muro A, Feliu-Soler A, Castellà J. Psychological impact of COVID-19 lockdowns among adult women: the predictive role of individual differences and lockdown duration. Women & health. 2021;61(7):668-79.

7. Daoud N, Berger-Polsky A, Abu-Kaf S, Sagy S. Sense of coherence among Bedouin women in polygamous marriages compared to women in monogamous marriages. Women & Health. 2020;60(1):43-59.

8. Antonovsky A. Unraveling the mystery of health: How people manage stress and stay well: Jossey-bass; 1987.

9. Hostinar CE, Miller GE. Protective factors for youth confronting economic hardship: Current challenges and future avenues in resilience research. American Psychologist. 2019;74(6):641.

10. Kanewischer E, Mueller C, Pylkkanen M, Tunks S. Hardships & Resilience: Families in a Pandemic. The Family Journal. 2021:10664807211054182.

---

## [Editor Report · Decision Letter 1]

17 Jan 2023

Trends in maternal mental health during the COVID-19 Pandemic – Evidence from Zambia

PONE-D-22-29001R1

Dear Dr. Falgas-Bague,

We’re pleased to inform you that your manuscript has been judged scientifically suitable for publication and will be formally accepted for publication once it meets all outstanding technical requirements.

Kind regards,

Ankit Jain, M.D.

Academic Editor

PLOS ONE
---

## [Editor Report · Acceptance letter]

25 Jan 2023

PONE-D-22-29001R1 

Trends in maternal mental health during the COVID-19 pandemic – Evidence from Zambia 

Dear Dr. Falgas-Bague:

I'm pleased to inform you that your manuscript has been deemed suitable for publication in PLOS ONE. Congratulations! Your manuscript is now with our production department. 

Kind regards, 

on behalf of

Dr. Ankit Jain 

Academic Editor

PLOS ONE